

# Mining Google Trends data for nowcasting and forecasting colorectal cancer (CRC) prevalence

Cristiana Tudor and Robert Aurelian Sova

Bucharest University of Economic Studies, Bucharest, Romania

## ABSTRACT

**Background:** Colorectal cancer (CRC) is the third most prevalent and second most lethal form of cancer in the world. Consequently, CRC cancer prevalence projections are essential for assessing the future burden of the disease, planning resource allocation, and developing service delivery strategies, as well as for grasping the shifting environment of cancer risk factors. However, unlike cancer incidence and mortality rates, national and international agencies do not routinely issue projections for cancer prevalence. Moreover, the limited or even nonexistent cancer statistics for large portions of the world, along with the high heterogeneity among world nations, further complicate the task of producing timely and accurate CRC prevalence projections. In this situation, population interest, as shown by Internet searches, can be very important for improving cancer statistics and, in the long run, for helping cancer research.

**Methods:** This study aims to model, nowcast and forecast the CRC prevalence at the global level using a three-step framework that incorporates three well-established univariate statistical and machine-learning models. First, data mining is performed to evaluate the relevancy of Google Trends (GT) data as a surrogate for the number of CRC survivors. The results demonstrate that population web-search interest in the term "colonoscopy" is the most reliable indicator to nowcast CRC disease prevalence. Then, various statistical and machine-learning models, including ARIMA, ETS, and FNNAR, are trained and tested using relevant GT time series. Finally, the updated monthly query series spanning 2004–2022 and the best forecasting model in terms of out-of-sample forecasting ability (*i.e.*, the neural network autoregression) are utilized to generate point forecasts up to 2025.

**Results:** Results show that the number of people with colorectal cancer will continue to rise over the next 24 months. This in turn emphasizes the urgency for public policies aimed at reducing the population's exposure to the principal modifiable risk factors, such as lifestyle and nutrition. In addition, given the major drop in population interest in CRC during the first wave of the COVID-19 pandemic, the findings suggest that public health authorities should implement measures to increase cancer screening rates during pandemics. This in turn would deliver positive externalities, including the mitigation of the global burden and the enhancement of the quality of official statistics.

Corresponding author
Cristiana Tudor,
cristiana.tudor@net.ase.ro

## INTRODUCTION

As of 2020, the *World Health Organization (WHO) (2022)* estimated that cancer remained the top cause of death globally, accounting for nearly 10 million fatalities (*Tudor, 2022a*). Yet, cancer is not just a major public health concern but also an economic issue whose burden is growing (*Cancer Atlas, 2022*). This is especially true for low- and middle-income nations, which accounted for 70% of cancer deaths recorded at the global level in 2020 (*American Cancer Society, 2022b*). Unsurprisingly, the UN Sustainable Development Goals (SDGs) (*United Nations, 2022*) include SDG target 3.4, which aims to "by 2030," reduce by one-third premature mortality from non-communicable diseases (NCDs) through prevention and treatment and promote mental health and well-being (*Kocarnik et al., 2022*; *Tran et al., 2022*) including the non-communicable disease of cancer.

Colorectal cancer (CRC) is the third most prevalent and second deadliest form of cancer worldwide (*Cervantes et al., 2022*; *Kadakuntla et al., 2021*; *Keum & Giovannucci, 2019*; *Mazidimoradi, Tiznobaik & Salehiniya, 2022*; *Uhlig et al., 2021*; *Xie, Chen & Fang, 2020*; *Xi & Xu, 2021*). Specifically, with 1,9 million new cases in 2020, colorectal cancer ranks third among all diagnosed cancers and accounts for 10.7% of all new cases, as shown in Table 1.

In 2020, colorectal cancer was the third most common cancer in the world for men, accounting for 11.4% of all new cases (*World Cancer Research Fund International (WCRF), 2022*). In contrast, CRC cancer was the second most prevalent cancer worldwide for women in 2020, accounting for 9.9% of new cases (WCRF, 2022). In addition, prior research has shown that CRC is more prevalent in advanced economies, while its incidence is increasing in middle- and low-income nations due to westernization, and early-onset CRC is also becoming more prevalent (*Xi & Xu, 2021*). In addition, new figures reveal that CRC is the second-leading cause of cancer-related mortality worldwide, accounting for nearly 1 million fatalities annually (*International Agency for Research on Cancer (IARC), 2022*).

Effectively addressing this increasing burden globally requires both a thorough understanding and accurate forecasting of its trends. Different from cancer incidence, cancer prevalence is the proportion of individuals within a population who have been diagnosed with cancer at some point in their lives, regardless of the date of diagnosis (*Maddams et al., 2009*). Compared to the average population, this population category, often dubbed 'cancer survivors', places a higher strain on the healthcare system (*Capocaccia et al., 2002*). In this context, cancer prevalence projections are key to assessing the future burden of the disease, efficiently planning resource allocation and creating service delivery strategies, and comprehending the changing environment of cancer risk factors (*Bray et al., 2013*; *Maddams, Utley & Møller, 2012*; *Smittenaar et al., 2016*). These predictions offer valuable intelligence to both the planners of health service resource allotting bodies and to providers of dedicated care and treatment, but, different from cancer incidence and mortality rates, cancer prevalence projections are not routinely issued by national and international agencies, and thus are not widely available (*Maddams, Utley & Møller, 2012*). The limited or non-existent cancer statistics for large parts of the world, together with the high heterogeneity encountered among world countries (*Thun*

| Table 1 Cancer incidence (worldwide, 2020). | | |
|---|---|---|
| Rank | Cancer | New cases (both sexes) |
| 1 | Breast | 2,261,419 |
| 2 | Lung | 2,206,771 |
| 3 | Colorectal | 1,931,590 |
| Rank | Cancer | New cases (Men) |
| 1 | Lung | 1,435,943 |
| 2 | Prostate | 1,414,259 |
| 3 | Colorectal | 1,065,960 |
| Rank | Cancer | New cases (Women) |
| 1 | Breast | 2,261,419 |
| 2 | Colorectal | 865,630 |
| 3 | Lung | 770,828 |

Note:
Data source: (World Cancer Research Fund International (WCRF), 2022: https://www.wcrf.org/cancer-trends/worldwide-cancer-data/).

*et al., 2010*), further complicate this task. All these factors further contribute to highlighting the importance and need for more research offering accurate cancer prevalence projections at various levels and form the motivators for the current study.

Time series forecasting is a critical and rapidly expanding topic (*Aras & Kocakoç, 2016*), but it remains a challenging task (*Petropoulos & Spiliotis, 2021*) that is further hindered by the aforementioned issues when it comes to producing cancer-related projections. However, to overcome the main pitfalls in cancer prevalence forecasting, over the last decade, the usefulness of Internet-search data for health informatics has been increasingly documented, with Internet resources becoming more accessible and providing data that can be used to assess and forecast human behavior (*Polgreen et al., 2008*; *Mavragani & Ochoa, 2019*; *Szilagyi et al., 2021*). Consequently, big data provided by the Google Trends platform has proven valuable in health and medical research (*Nuti et al., 2014*; *Szilagyi et al., 2021*; *Tudor & Sova, 2022*), first in infodemiology studies (*Eysenbach, 2011*; *Bernardo et al., 2013*; *Mavragani & Ochoa, 2019*; *Mavragani, 2020*; *Kamiński, Łoniewski & Marlicz, 2020*), but lately being increasingly employed in cancer-related research (*Schootman et al., 2015*; *Greiner et al., 2022*; *Tudor, 2022a*). Consequently, as per *Salathé et al. (2012)* and *Sulyok, Ferenci & Walker (2021)*, among others, online digital data sources have been proven to improve disease surveillance, monitoring, modeling, and forecasting.

Moreover, the COVID-19 pandemic and the related measures imposed by governments worldwide to tackle it have had an impact on all aspects of life, including medical care (*Eftimov et al., 2020*; *Trinh et al., 2022*). As such, a non-trivial consequence of the pandemic-related restrictions consists of delays in cancer screenings and early diagnostics (*Sharpless, 2020*; *Bakouny et al., 2021*; *Greiner et al., 2022*; *Trinh et al., 2022*), halting of clinical trials, and delays in treatments (*Bakouny et al., 2020*; *Richards et al., 2020*), thus decreasing the relevancy of official statistics and having negative consequences for cancer projections and overall for cancer research (*Saini et al., 2020*). In turn, this further

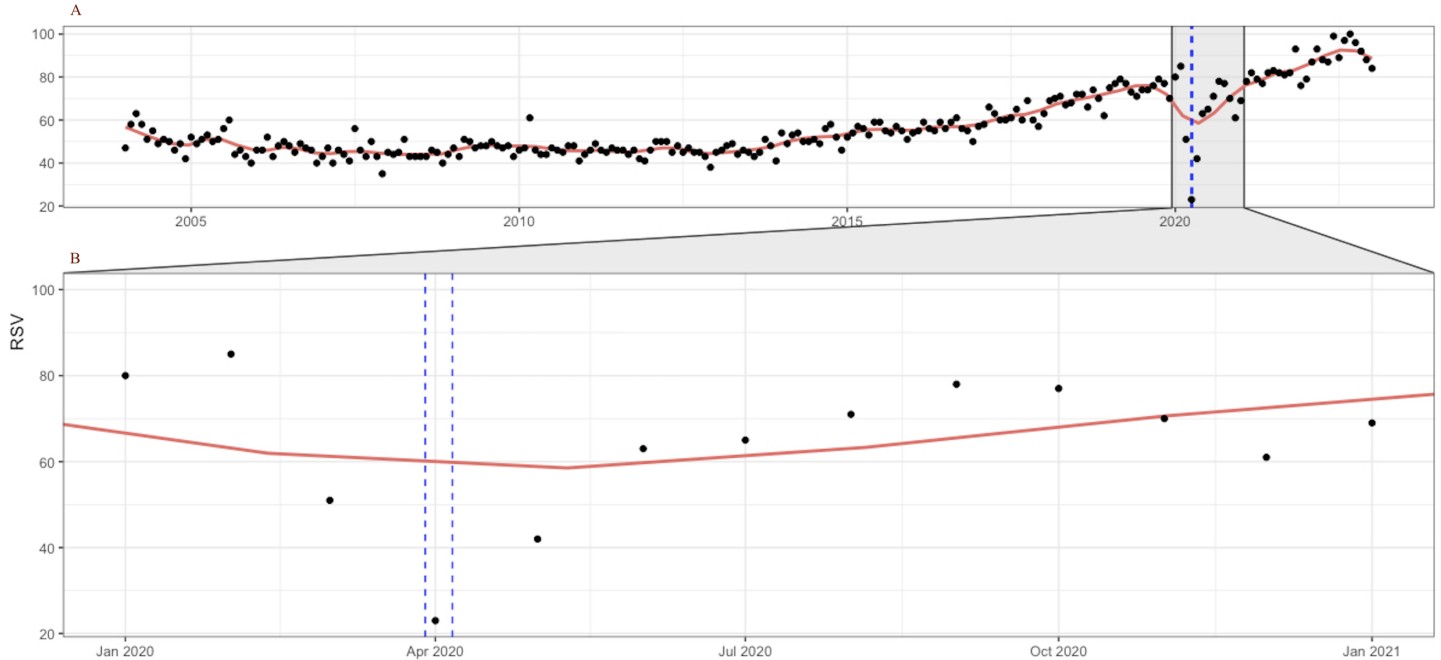

**Figure 1 Google Trends search index for the keyword "colonoscopy" (January 2004–December 2022, category = "Health") (A); Zoom in on January–December 2020 evolution (B).** LOESS smoothing shown in red. Source: authors' representation in R software (*Wickham et al., 2023*).

increases the utility of public interest in cancer as reflected in web-search data to substitute for cancer indicators that have been affected by the coronavirus pandemic.

For example, Fig. 1 shows the all-time evolution of population interest for the keyword "coloscopy" at the global level and identifies without a doubt the plummet occurring during the month of April 2020, when the lowest popularity of the term has registered. In the following months, the trend reversed and continued on an ascending trend thereafter, mirroring the evolution of screening rates, as documented in the literature. The lower part of the chart contains a zoomed view over the entire year 2020, underlining the sharp decrease in interest starting in March 2020 and reaching an all-time low level in April 2020. The chart also draws a locally weighted polynomial regression line (LOESS) (*Cleveland, 1979*; *Cleveland & Devlin, 1988*) to increase trend readability.

Furthermore, the seasonal plot depicted in Fig. 2 contributes to clearly identify the particular pattern present for the year 2020 and the disruption that occurred in the relative search volume index (RSI) series in April 2020 during the first wave of the COVID-19 pandemic.

The data thus confirm previous findings that showed significant decreases in the number of screenings for a variety of cancers, including colorectal, immediately after the onset of the COVID-19 pandemic, which was no longer present by June. This further attests to the relevancy of the relative search index as a proxy for cancer screenings, with important implications for research, policy, and industry. The chart also helps to reveal other underlying seasonal patterns, such as a decrease in web-search interest for CRC in December each year, which should be considered for accurate forecasting.

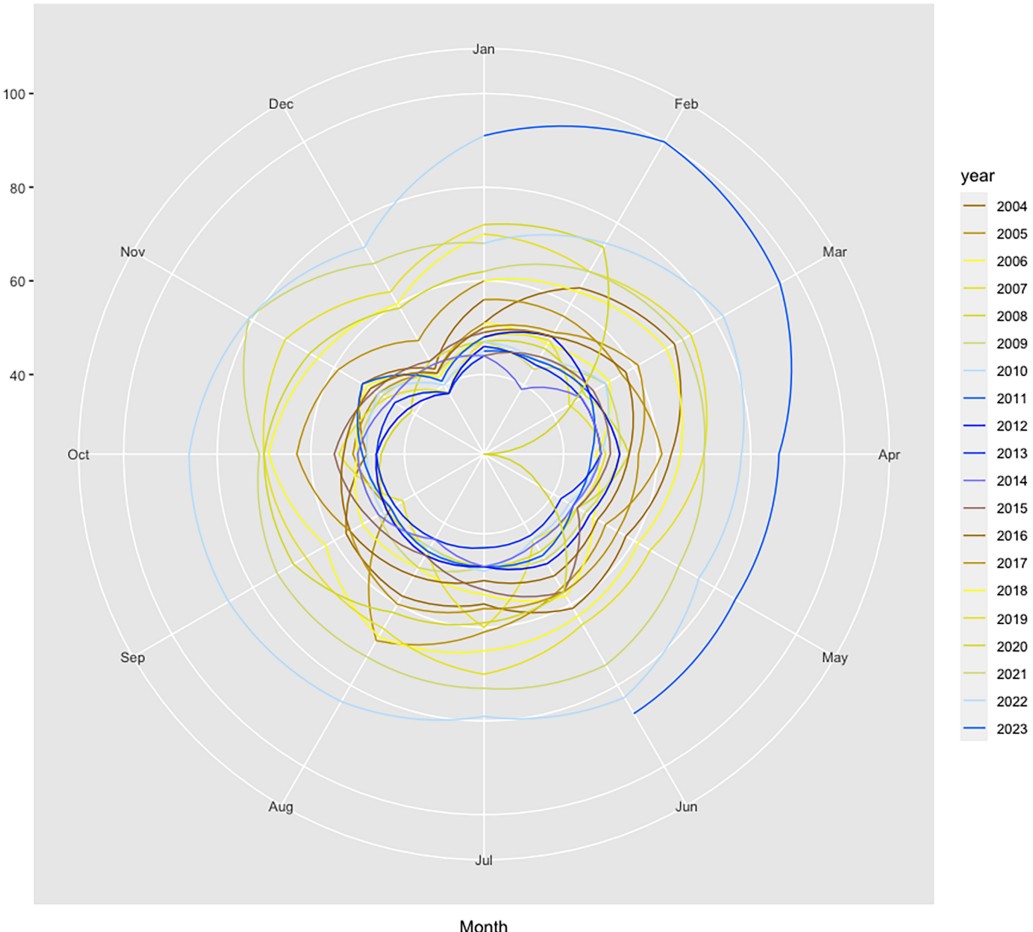

**Figure 2** **Polar seasonal plot for GT queries submitted for "colonoscopy" worldwide during 2004–2022.** Source: authors' representation in the "colorblindr" and ggplot2 R packages (*Wickham et al., 2023*).

Additionally, GT data has been shown to present non-linear characteristics that must be accounted for in modeling and forecasting endeavors. Very recently, *Borup & Schütte (2022)* demonstrate that taking them into account in models that employ GT data greatly increases performance over linear models. Amongst state-of-the-art models, artificial neural networks (ANNs) have been particularly useful for modeling complex systems due to their adaptability, parallel processing, and learning capabilities (*Sarangapani, 2018*), while properly accounting for data non-linearity (*Pasini, 2015*; *Tudor, 2022b*). There are many types of ANN architectures, amongst which the most widely used are feed-forward neural networks (FNNs) or multilayer perceptrons (MPs) (*Hsieh, 2004*).

The primary objective of the current study is to provide accurate forecasts for the CRC prevalence rate through a three-step forecasting framework that embeds an FNN autoregressive model (FNNAR). To reach its goal, it sources Google Trends data to assess public interest in CRC, and calibration is performed using the FNNAR model that captures non-linear patterns and traditional univariate forecasting techniques, including the exponential smoothing state space model (ETS) (*Ord, Koehler & Snyder, 1997*; *Hyndman*

*et al., 2002*) and the auto-regressive integrated moving average model (ARIMA) developed by *Box & Jenkins (1970)*, and then the best performing forecasting model in an out-of-sample setting is identified and used to issue forecasts for the CRC prevalence rate by the year 2025. To the best of the authors' knowledge, this is the first attempt to model and forecast CRC prevalence rates by using GT web-query data. Two secondary goals are also proposed and followed: (i) to document trends and heterogeneity in the prevalence of colorectal cancer by regions and country income level; (ii) to assess the efficacy of using public interest, reflected by Google Trends data, as an indicator of actual cancer prevalence and develop a CRC prevalence model based on web-search data. Different from the thin body of literature preoccupied with modeling and forecasting cancer indicators, the current research makes the following contributions: (i) it focuses on prevalence rates, as opposed to screening or incidence rates; (ii) it documents the capability of GT data to proxy for cancer statistics through multiple tools and assesses the relationship between CRC-related web searches and CRC prevalence statistics to identify the best keyword capable of proxying or even substituting prevalence rates. (iii) it performs preliminary testing and comparatively explores the in-sample and out-of-sample capabilities of alternative models, both statistical and machine-learning (*i.e.*, as per the categories proposed by *Breiman (2001)* to confirm the over-performant model and its optimum parameters.

The manuscript is structured as follows: The second section describes the data employed in the investigations and introduces the various elements embedded in the forecasting framework. The next section presents the findings, whereas the fourth section contains a discussion of the main results, and the fifth section concludes the study.

## MATERIALS AND METHODS

### Data

CRC prevalence data were extracted from the publicly available database of the Our World in Data (OWID) platform *via* the owid() function embedded in the dedicated R software package (https://ourworldindata.org/). At its turn, OWID sources CRC prevalence data from the Global Burden of Disease (GBD) database of the Institute for Health Metrics and Evaluation (IHME) (https://www.healthdata.org/gbd), which is thus the original cancer statistics source. The CRC prevalence data covers the 2004–2019 period (*i.e.*, T = 16) for a total of 202 individual countries and country panels. First, heterogeneity across world countries is assessed through "plotmeans" style plots with the R's "gplots" package of *Gregory, Warnes & Lodewijk (2016)*. This method carries the non-trivial advantage of plotting group means along with confidence intervals (CI), which, together with descriptive statistics reported in Table 3, offers an insightful view of the extent of heterogeneity present in the data. Of note, the *t* distribution is used by default to compute confidence intervals within the plotmeans() function. However, as *Rovetta (2023)* explains, confidence (or credible) intervals concern the mean distribution and generally are not the best indicators for the dataset variability.

**Table 2 Income and geographical panels employed in the study.**

| Income panels | Geographical panels | Abbreviation |
|---|---|---|
| Higher income | Sub-Saharan Africa | SSA |
| Upper middle income | South Asia | SA |
| Lower middle income | North America | NAm |
| Low income | Middle East & North Africa | MENA |
| | Latin America & Caribbean | LAC |
| | Europe & Central Asia | ECA |
| | East Asia & Pacific | EAP |

Moreover, the heterogeneity in worldwide CRC prevalence is further explored by employing both income-based and geography-based panels, as per the Word Bank (WB) classification.

This approach is particularly helpful for informing global equitable health policies. Consequently, four income-based panels and their corresponding CRC prevalence rates are extracted. Subsequently, seven geography-based panels are also delineated, and the relevant observations are sourced. Table 2 reflects the income and geographical panels extracted from OWID, along with the abbreviations employed for the latter category.

The Google Trends platform (https://www.google.com/trends/) reports the relative search volume index (RSI) of a keyword in a region, over a specified time period, with an RSI of 100 reflecting the point of peak popularity (*Silva et al., 2019*). As such, the RSI is estimated as per Eq. (1) and subsequently scaled from 0 to 100 (*Tudor & Sova, 2022*).

$$RSI = \frac{Total\ search\ volume\ for\ "keyword"\ submitted\ in\ the\ specified\ geography\ over\ the\ specified\ period}{Total\ searches\ submitted\ in\ the\ specified\ geography\ over\ the\ specified\ period} \quad (1)$$

Google Trends data is sourced in this study by making use of R's "gtrendsR" package (*Massicotte & Eddelbuettel, 2022*), calling the gtrends() function, specifying alternatively "colorectal cancer" and "colonoscopy" as the search terms of interest, the default geography of "all" corresponding to global queries, and specifying the "health" category (*i.e.* category no. 45) to assure reliable results. Moreover, the string specifying the time span of the query is set to "all," which implies that monthly RSI data starting on January 2004 (*i.e.*, the beginning of Google Trends) and spanning to December 2022 is extracted. Consequently, a time series of 229 monthly RSI observations is available and will be employed for modeling and forecasting purposes in this study. *Tudor (2022a)* offers further relevant details on the methodology behind the RSI index. The "gtrendsR" package is not only a great tool for extracting GT data, but allows the specification of multiple relevant elements, is capable of performing an interest-by-region analysis and is useful for making searches reproducible (however, as explained later, sample instability is an issue that must be accounted for). It should be mentioned that the ISO language code only influences the data returned by related topics, as per the package description, and thus does not impact the sourced data.

## Method

This study develops a three-step framework to forecast monthly CRC prevalence evolution based on Google Trends relative search index.

First, a monthly model of CRC prevalence is estimated based on RSI for the world panel (as per World Bank), such that:

$$CRC = f(RSI) \tag{2}$$

where the CRC prevalence in year $i$ is modelled as a linear function $f$ of the same-year RSI, plus some white noise.

To perform the estimation, the monthly RSI index is first converted to annual averages to match the frequency of the CRC prevalence series imported from the OWID database. Moreover, the parameters are estimated *via* the ordinary least squares (OLS) estimator. Of note, the main research interest does not lie on fine-tuning the CRC-RSI relationship, but rather on the second step of the framework, *i.e.*, accurate forecasting of RSI. Additionally, particular care is given to the choice of the estimation period, juggling the need to include enough observations and the fact that the Google platform has taken off as a main search engine only in the most recent decade. As per Britannica (https://www.britannica.com/topic/Google-Inc), by 2004, Google was processing daily 200 million searches, increasing to over 3 billion by the end of 2011. As of March 2023, Google's market share of the global search market across all devices is nearly 84%, making it the dominant search engine worldwide (*Statista, 2023*), which translates into 9 billion daily searches (*EarthWeb, 2023*). In light of these figures and aiming to increase the relevancy of current findings, the estimations performed within the first step of the framework (*i.e.*, to assess the link between distinct RSI pertaining to alternative relevant keywords and official CRC prevalence statistics), have been limited to the most recent decade of available data. However, this in turn reduces the number of observations in the data sample employed in the first step within the proposed framework.

Second, the RSI time series is modeled in-sample, tested out-of-sample, and ultimately forecasted and validated against official statistics and/or results of other studies, where available. To this end, multiple tools are employed.

As such, the nonlinearity of monthly RSI data is examined by two alternative tests: the Tsay test for quadratic nonlinearity (*Tsay, 1986*), as well as the likelihood ratio test for threshold autoregression (LR) proposed by *Chan (1991)*. In both cases, the null hypothesis is that the time series includes an AR process (*Munim, 2022*). Next, for modeling and forecasting purposes, the RSI index is split into a training set (made up of 80% of observations) used for in-sample fitting of three alternative models (*i.e.*, ETS, ARIMA, and FNNAR) and a testing set (containing the remaining 20% of the observations) on which the forecasting ability of the best-fit model in-sample in each category (*i.e.*, best ETS, best ARIMA, best FNNAR) is comparatively assessed. For this purpose, several forecasting accuracy statistics (*Hyndman & Athanasopoulos, 2018*) are calculated (shown below). Moreover, the Diebold and Mariano (DM) test for superior forecasting ability (*Diebold & Mariano, 1995*) is estimated to confirm the superior forecasting ability of the best forecasting model relative to the second ranked candidate.

Root mean squared error:

$$RMSE = \sqrt{mean(e_t^2)} \tag{3}$$

Mean absolute error:

$$MAE = mean(|e_t|) \tag{4}$$

Mean absolute percentage error:

$$MAPE = mean(|p_t|) \tag{5}$$
$$\text{where } p_t = \frac{100e_t}{y_t}$$

Mean absolute scaled error:

$$MASE = mean(|q_j|), \tag{6}$$

$$\text{where } q_j = \frac{e_t}{\frac{1}{N-1}\sum_{i=2}^{N}|y_t - y_{t-1}|} \text{ in the case of non-seasonal series and}$$

$$q_j = \frac{e_t}{\frac{1}{N-m}\sum_{i=m+1}^{N}|y_t - y_{t-m}|} \text{ when the time series is seasonal.}$$

In all cases, the forecast error is defined as:

$$e_{T+h} = Y_{T+h} - \hat{Y}_{T+h|T} \tag{7}$$

To sum up, the data splitting rule for the training-testing RSI modeling and forecasting is given in the script snippet in Box 1.

Box 1. Script for implementing the splitting rule

```
RSI <- trends$interest_over_time
x<- ts(RSI$hits)
test_x <- window(x, start=c(185,1),end=c(229,1))#testing window
x <- window(x, end=c(184,1)) #training window
```

Simple statistical methods have dominated the field of time series forecasting for many decades, both in academia and industry, due to their relatively accurate, fast-to-compute, and interpretable forecasts, with neural networks emerging as especially accurate in univariate time series forecasting (*Semenoglou, Spiliotis & Assimakopoulos, 2023*). However, due to their capability of revealing the influence of many parameters, multivariate time series models have uncontested advantages, but their goodness of fit is intrinsically related to the quality and availability of data related to these parameters (*Atchadé & Sokadjo, 2022*). Thus, given that the more complex the model, the more the need for data, and the lesser the availability and quality of cancer prevalence statistics, especially in less-developed economies and during the COVID-19 pandemic, we rely on the best-identified univariate model in the current research. Our choice is further backed by the findings of *Ziel & Weron (2018)* that conclude that, for the

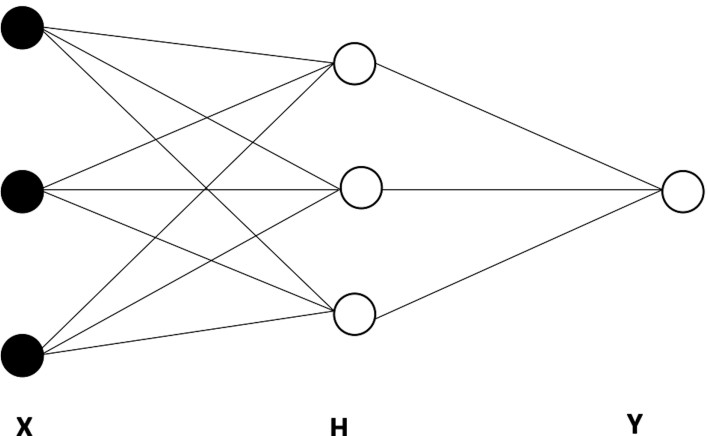

**Figure 3 The basic architecture of an artificial neural network (ANN), including one hidden layer with three nodes.**

task of electricity price forecasting, the multivariate modeling approach does not uniformly outperform the univariate one across 12 distinct electricity spot price datasets and is even often outperformed by the latter. Furthermore, we rely on the findings of *Jaidka et al. (2021)* that study 208 Designated Market Areas (DMAs) throughout the United States and find that information-seeking behavior from Google Trends data is 19% more accurate than sociodemographic and regional controls at predicting the prevalence of noncommunicable diseases. Consequently, as per *Ziel, Steinert & Husmann (2015)*, we argue that our approach can surpass the complexity of cancer prevalence forecasting by offering several advantages relative to multivariate settings, such as being capable to model the search volume index and to provide accurate forecasts without the need for any data manipulation nor for any additional information, and, not in the least, by employing efficient and rapid state of the art estimation techniques.

As mentioned earlier, artificial neural networks (ANNs) have emerged as particularly useful for modeling and forecasting complex systems while accounting for non-linearities. The FNNs, are the most extensively utilized neural network (NN) models (*Hsieh, 2004*). A neural network is made up of several small computational units or nodes that are stacked in layers and run in parallel, the, connections among nodes, and an activation function (*Chen & Billings, 1992*; *Allende, Moraga & Salas, 2002*). Figure 3 provides the basic architecture of an ANN, which is made up of an input layer X, linked to at least one hidden H, which is, in turn, linked to the output layer Y (*Chen & Billings, 1992*).

The feedforward network is a type of neural network in which the input feeds forward through the network layers to the output. In the case of a feedforward neural network autoregression (FNNAR) model, the input layer consists of the past observations of a series (*Munim, Shakil & Alon, 2019*). Consequently, the vector of predicted values Y is found as follows:

$$Y = f(H) \tag{8}$$

where

$$H = \{\text{weight matrix } [(p * k)] * X + \text{bias vector} \qquad (9)$$

and $f$ is the activation function, with $p$ standing for the nonseasonal inputs for the linear AR process, and $k$ for the number of nodes in the hidden layer.

FNNAR models are fitted through automated algorithms in the "forecast" package in R software. In particular, the nnetar() function makes 25 repetitions and finds parameters through AIC minimization. By default, the number of nodes in the hidden layer is calculated as $k = (p + P + 1)/2$.

Additionally, to assure the robustness of findings, ETS and ARIMA models are automatically fitted in the R environment.

The exponential smoothing state space model (ETS) is based on the work of *Holt (1957)*, *Winters (1960)*, and *Brown (1959)*. Predictions given by ETS are weighted averages of prior data, with greater weights assigned to more recent observations and weights decreasing exponentially (*Hyndman & Athanasopoulos, 2018*; *Silva et al., 2019*). ETS estimates comprise three components: the trend (T), seasonal (S), and error (E) components. In addition, the trend includes both a level term (l) and a growth term (b). According to *Yang et al. (2015)*, the trend and seasonal components might be none (N), additive (A), additive damped (Ad), multiplicative (M), or multiplicative damped (Md). Hence, an ETS model is represented by a three-character string (Z,Z,Z) (*Perone, 2021*), with the first Z representing the error assumption of the state-space model and the second and third Zs representing the trend type and the season.

The use of the automated ETS technique *via* the ets() function inside the forecast package in R (*Hyndman & Khandakar, 2008*; *Hyndman et al., 2022*) assesses 30 ETS equations during the modeling procedure *i.e.*, all 30 formulae are provided by *Hyndman & Khandakar (2008)*.

The approach optimizes the smoothing parameters and the starting state variable and employs a penalized likelihood, *i.e.*, the corrected Akaike's Information Criterion (AICc), to choose the best model on the training set, which is then used to generate point predictions.

Autoregressive integrated moving average (ARIMA) models were created by *Box & Jenkins (1970)* and are among the most widely used parametric time series analysis and forecasting methods (*Silva et al., 2019*).

In equation form, according to *Hyndman & Athanasopoulos (2018)*, an ARIMA (p,q,d) (P,Q,D) seasonal model is given as:

$$(1 - \varphi_1 B - \ldots - \varphi_p B^p)(1 - \Phi_1 B^s - \ldots - \Phi_P B^{sP})(1 - B)^d (1 - B^s)^D Y_t$$
$$= (1 - \theta_1 B - \ldots - \theta_q B^q)(1 - \Theta_1 B^s - \ldots - \Theta_P B^{sQ})\varepsilon_t \qquad (10)$$

where s is the seasonal period, the lowercase and the capital letters represent nonseasonal and seasonal parameters, whereas $\varepsilon_t$ is a zero-mean random variable with the standard deviation $\sigma$. The "auto.arima" function from the "forecast" package in R software conducts ARIMA modeling using an automated and optimized method that is capable of efficiently traversing the space of models in order to identify the ideal model. The function determines if seasonal differencing is required for the training data, calculates unit root

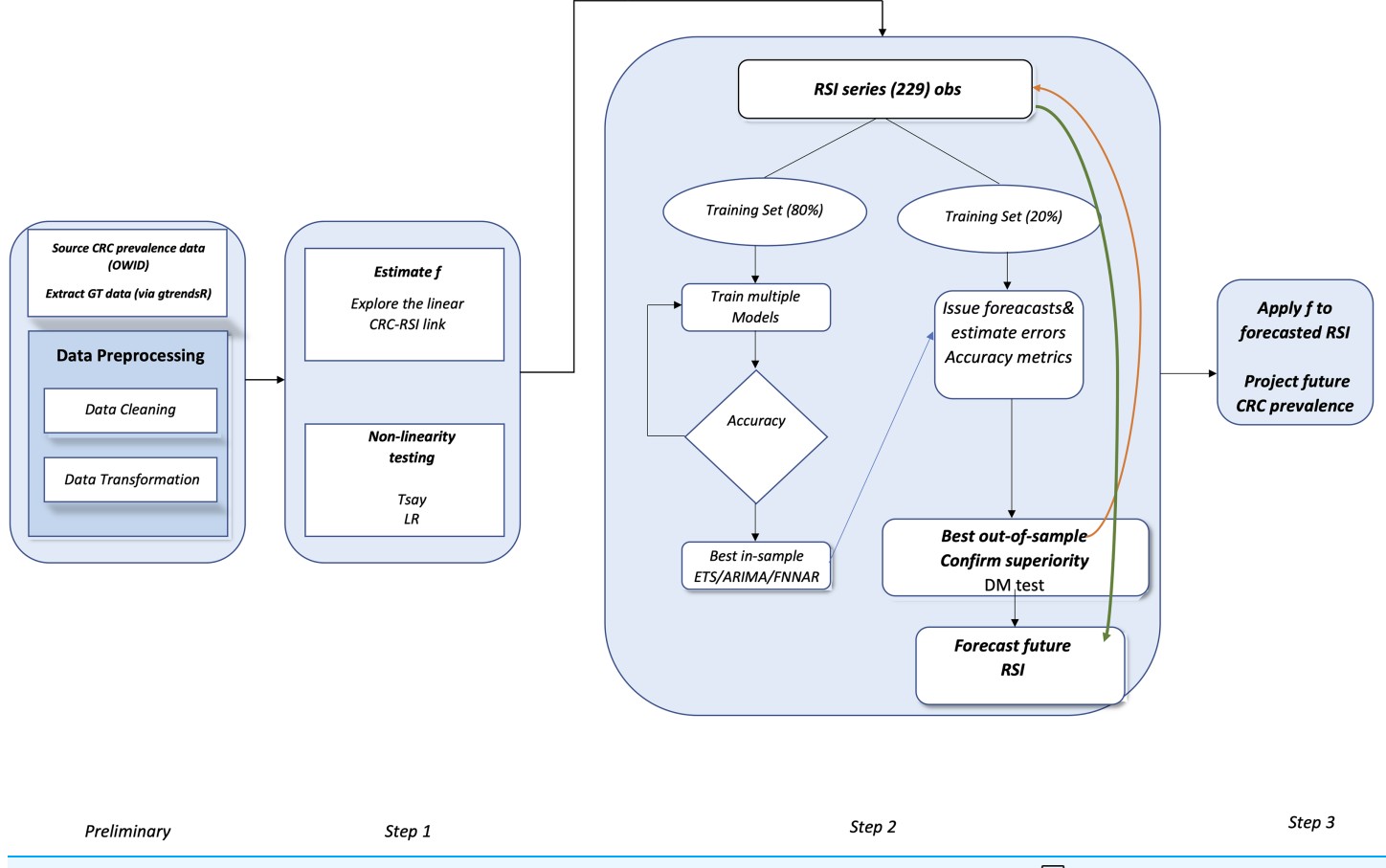

**Figure 4** Flowchart of the proposed method.               

tests, and selects model parameters using a step-by-step AICc reduction procedure (*Tudor, 2022a*, *2022b*).

The best forecasting model on the out-of-sample set is used to issue point forecast for the RSI index setting a forecasting horizon $h$ of 24 months, thus reaching the end of 2024. Of note, $h$ is chosen by complying with two important criteria in time series forecasting. First, ideally the forecasting horizon must be shorter than the testing window (*Hyndman & Athanasopoulos, 2018*). Second, longer-term forecasts have been shown to fail do provide superior information relative to the mean of the explained variable (*Breitung & Knüppel, 2021*). Consequently, setting the length of $h$ equal to 24 assures that both criteria are met.

Third, the function $f$ estimated at step one is applied to the RSI series forecasted at step two, under the no-change hypothesis, in order to produce forecasts for the CRC prevalence rate up to the end of 2024, as follows:

$$\widehat{CRC} = \hat{f}\left(\widehat{RSI}\right) \tag{11}$$

To sum-up, the proposed framework embeds the sequential main steps depicted in Fig. 4.

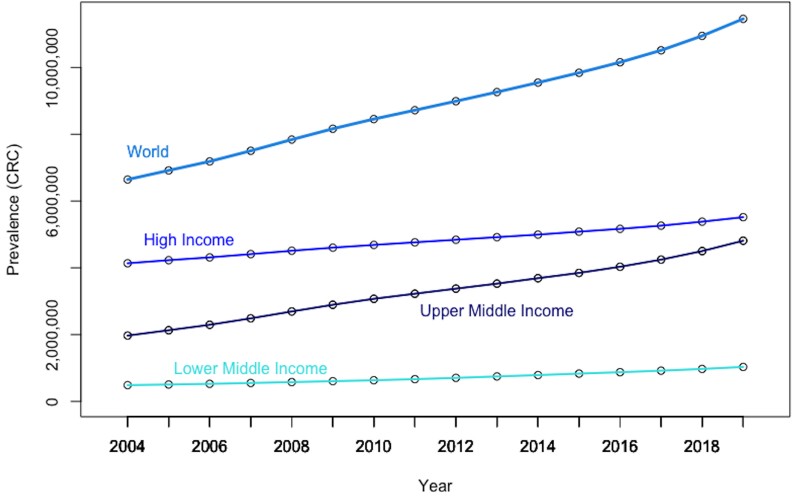

**Figure 5 Evolution in CRC prevalence rates at the world level and by income-based country panels.**
Authors' representation in R software. Our World in Data (OWID) https://ourworldindata.org/. The
low-income panel, showing a flat line at zero on the current scale, is eliminated from the chart to improve
overall readability. Authors' representation in R software. Source of data, OWID.

## RESULTS

### Worldwide trends and heterogeneity in CRC prevalence

Figure 5 reveals that the global panel presents a rapidly increasing CRC prevalence rate
over the last decades, which is mostly driven by the upper-middle income panel,
confirming that the rising number of CRC cases is creating an increasing global public
health concern (*Xi & Xu, 2021*).

The graphical depiction further shows that the prevalence rate of CRC in high-income
countries (*i.e.*, as per the World Bank classification) has consistently been about 10 times
that of low- and middle-income economies. Moreover, it should be noted that the growth
rate of CRC prevalence is significantly steeper in upper-middle income countries than in
both rich and poor countries, suggesting increased exposure to CRC risk factors, albeit at a
high basal level. Nevertheless, it is increasing rapidly in less developed countries due to
increased exposure to CRC risk factors.

Furthermore, there is also high geographical heterogeneity in CRC prevalence at the
world level (Fig. 6). Thus, data indicates that the East-Asia& Pacific region is registering
both a high basal CRC prevalence level, as well as a high growth rate, confirming that
immediate measures are paramount to offer some relief against this burden.

Additionally, the heterogeneity amongst individuals is also apparent in Fig. 7, which
denotes the mean CRC prevalence level over the sample period for each of the 202
countries included in the analysis. The plot is issued through the plotmeans() function in
the "gplots" package, which uses the *t* distribution to compute confidence intervals, with a
0.95 confidence level for error bars.

The high country heterogeneity also emerges from the descriptive statistics for the entire
sample, which are reported in Table 3. Estimations show a very high standard deviation

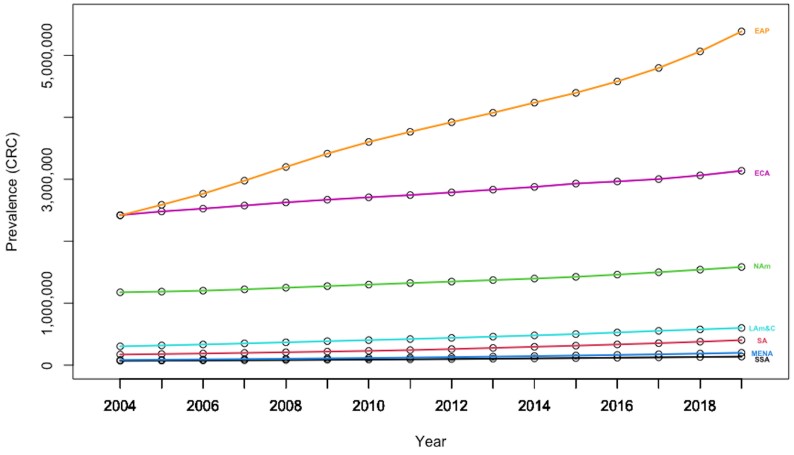

**Figure 6 Evolution in CRC prevalence rates by geography-based country panels.** Authors' representation in R software. Source of data: Our World in Data (OWID) https://ourworldindata.org/.

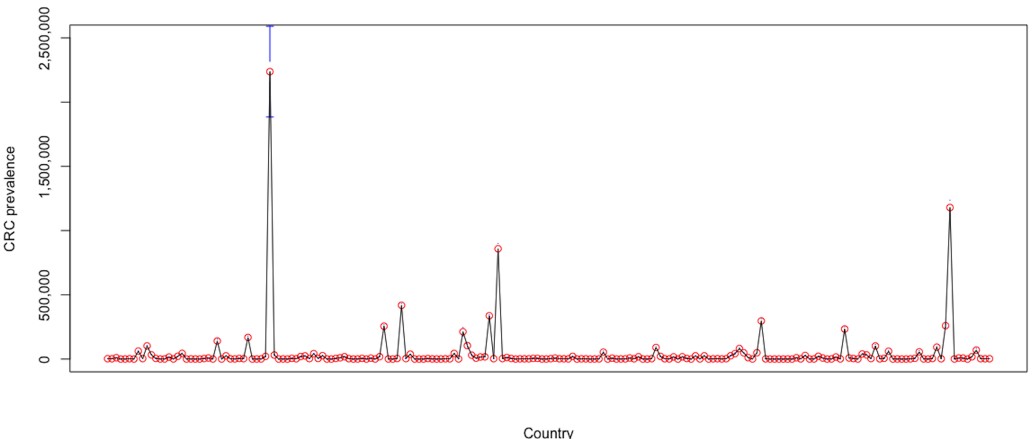

**Figure 7 Heterogeneity amongst countries over 2004–2019.** Authors' representation in R software. Source of data, Our World in Data (OWID) https://ourworldindata.org/.

**Table 3 Descriptive statistics for CRC prevalence during 2004–2019.**

| Statistic | CRC prevalence |
|---|---|
| Mean | 86,722.88 |
| Standard deviation | 657,663.06 |
| Min | 0.00 |
| Max | 11,457,627 |

(657,663.06) and huge range (11,457,627) for the levels of CRC prevalence for world countries during 2004–2019, whereas its mean level was 86,722.88.

## Relevancy of alternative RSIs for CRC prevalence

Figure 8, which plots the trends of the CRC prevalence and of the RSI "colonoscopy" in panel a, and the scatterplot between the two variables in panel b reveals a disaggregation

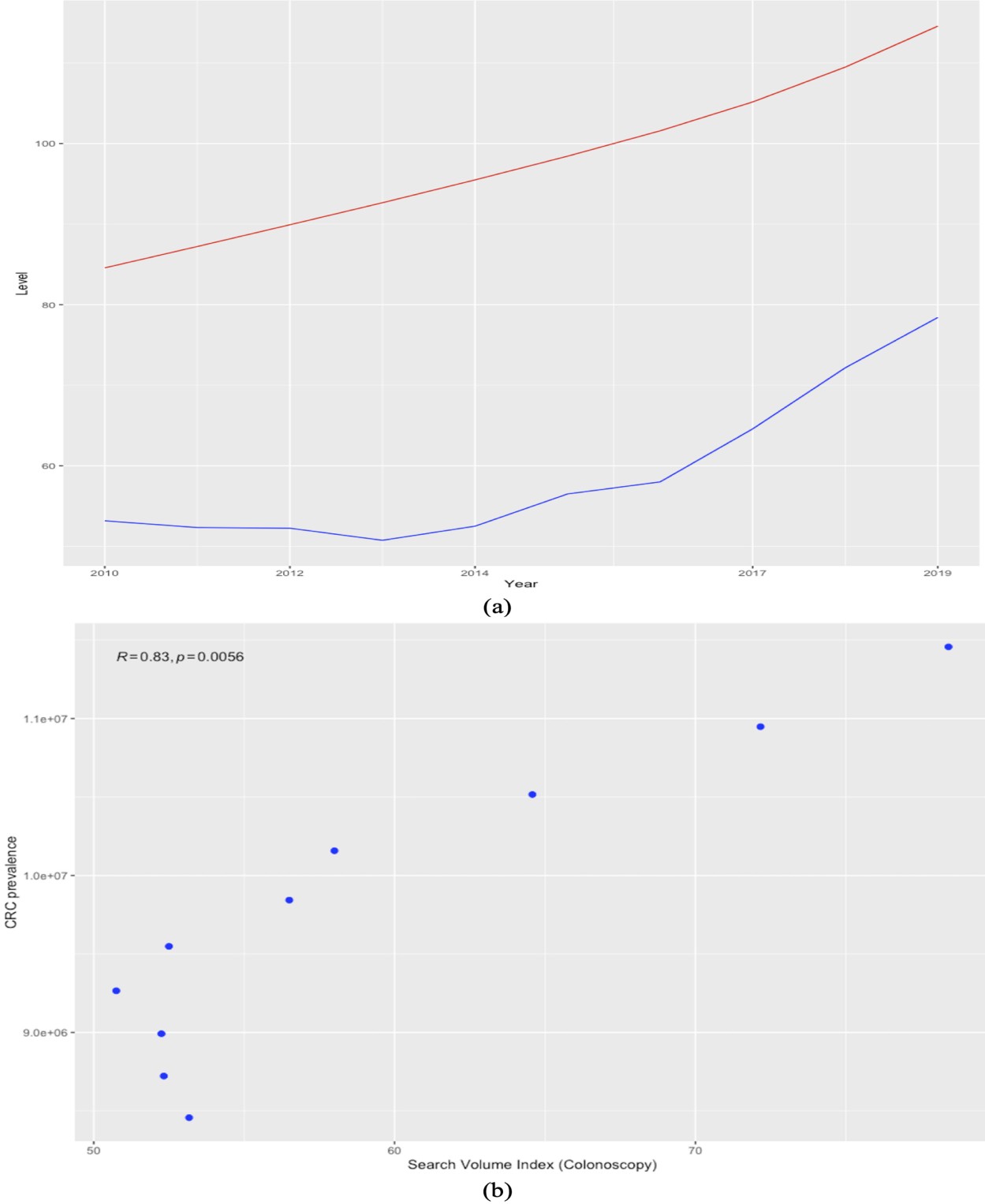

**Figure 8 Trends in worldwide cancer prevalence (×100,000, red line) and in the relative search index (RSI) for keyword "colonoscopy" (blue line) (2010–2019) (A). the relationship between the two variables over 2010-2019 (scatter plot) (B).** The Spearman correlation coefficient is automatically reported on the plot issued with the ggscatter() function in the "ggpubr" package.

between the two series during the first years in the sample, when the RSI series presents a decreasing trend, whereas the CRC prevalence series shows an uninterrupted increasing trend throughout the analysis period. This in turn leads next to the implementation of a joinpoint (or change-point) regression analysis (*Gillis & Edwards, 2019*) to confirm and assess the changing trends that emerge from the visual inspection. Of note, this kind of analysis has been routinely applied to detect changing trends in cancer mortality and/or incidence series (see, among others, *Qiu et al., 2009*; *Crispo et al., 2013*; *Sarakarn et al., 2017*; *Wilson, Bhatnagar & Townsend, 2017*). Results of the joint-point regression analysis, graphically reflected in Fig. 9, confirm that the RSI series presents one join point in 2014, leading to two periods with distinct trends, as follows: a negative trend until 2014, and an increasing trend during 2014–2019 (Fig. 9). On the other hand, the joinpoint regression analysis also found one joinpoint in CRC prevalence series around 2014, but in this case the joinpoint delineates two positive trends. All estimations have been performed R's software "segmented" package. Hence, given the disaggregation between the series during the first segment, to comply with the linearity assumption, we subset the secondary segment to implement the linear models and estimate the statistics of interest.

Estimation results for the two linear models are summarized in Table 4. Of note, as the RSI indexes are scaled from zero to 100 by construction, to make the data comparable and obtain more consistent estimates, the CRC prevalence rates are divided by 100,000 prior to estimating the model. First, it should be noted that effect sizes are the paramount outcome of empirical statistical research (*Lakens, 2013*). As explained by *Lakens (2013)*, which in turn cites *Rosenthal (1994)*, effect sizes are usually classified into two families: the d family (*i.e.*, standardized mean differences) and the r family (*i.e.*, measures of strength of association), whereas the d family effect sizes are conceptually determined by the difference between observations divided by the standard deviation of these observations, and the r family effect sizes characterize the proportion of variance that can be attributed to group membership. Moreover, as *Lee (2016)* explains, *p*-values can only inform of the statistical significance, but not the magnitude of the effect, whereas CIs mitigate, but do not solve, this issue by providing a range of possibility. Furthermore, it should be mentioned that the Shapiro-Wilk test (*Shapiro & Wilk, 1965*) estimated for the three series could not reject the null hypothesis that all samples stem from a normal distribution.

In light of these considerations, we estimate and report the Pearson correlation coefficient and the value of Cohen's d statistics (*Cohen, 1988*) together with its correction for sample bias (*i.e.*, known as Hedges' *g*) between the CRC prevalence series and each RSI series, including the 95% CIs for all estimates. Both Cohen's d and Hedges' *g* statistics are estimated by calling the cohen.d() function within the Efficient Effect Size Computation or "effsize" package (*Torchiano, 2020*) in R software. Results agree that the RSI for the keyword "colonoscopy" is the best indicator for cancer prevalence, showing a 95% CI for the Pearson correlation coefficient of (0.9324; 0.9992), and a goodness-of-fit (R-squared) of 98.56%, indicating that it can explain most of the variability in CRC prevalence at the world level during the most recent years of available data. In turn, the RSI for the keyword "colorectal cancer" can also be a reliable indicator for cancer prevalence, although not a substitute, contributing to explain close to 73% and showing lower correlation with the

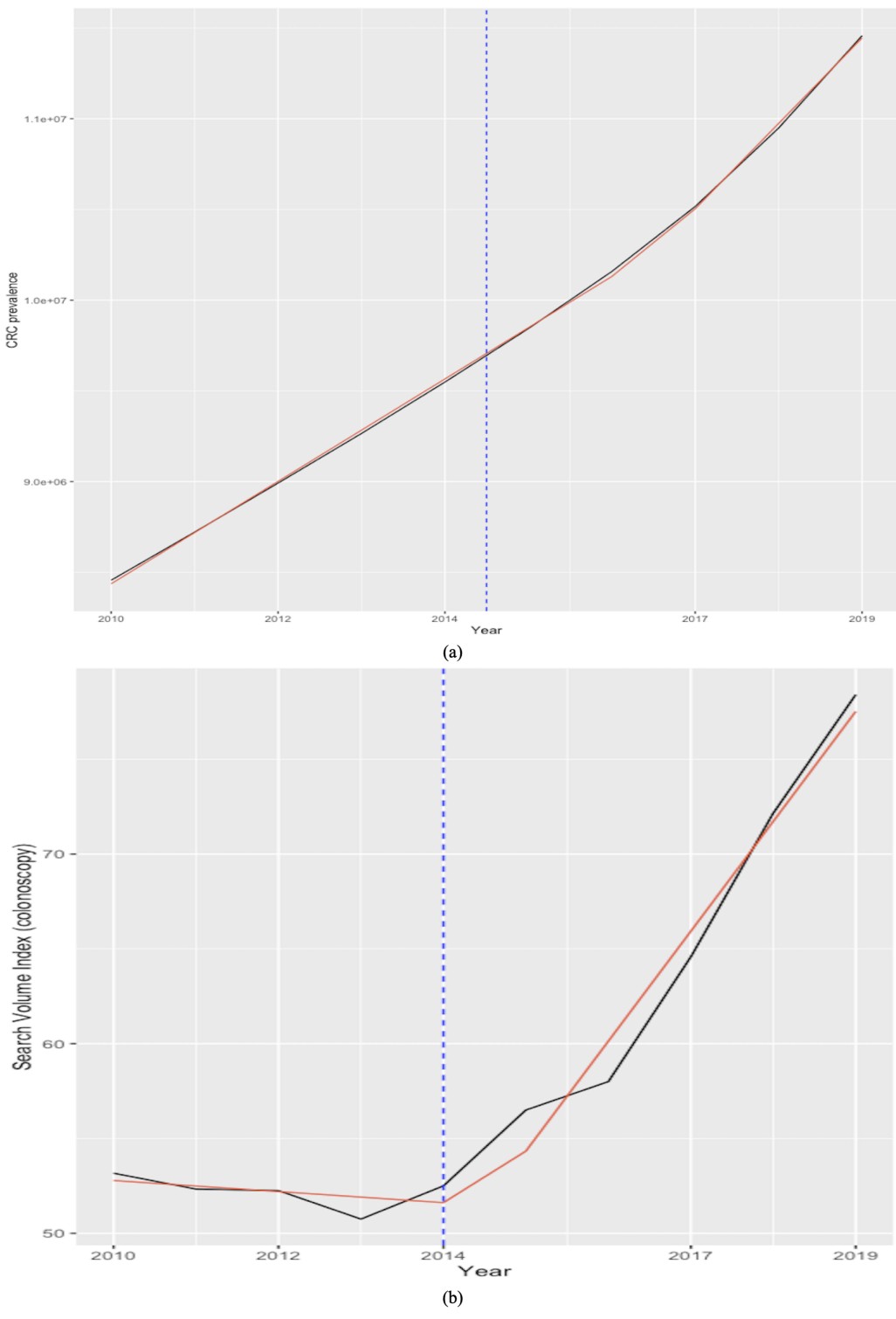

**Figure 9 Joinpoint regression analysis (CRC prevalence rate in A and RSI for keyword "colonoscopy" in B).** Source of data: CRC prevalence data is sourced from Our World in Data (OWID) https://ourworldindata.org/, which in turn sources data from IMHE, and RSI data is sourced from Google Trends. The "segmented" package within R software is used for the implementation of the joinpoint regression.

**Table 4 Estimates of the link between CRC prevalence rate and alternative RSI indexes.**

|  | RSI "colonoscopy" | RSI "colorectal cancer" |
|---|---|---|
| Pearson corr | 0.9927 | 0.8567 |
| (Lower 95%; Upper 95%) | (0.9324; 0.9992) | (0.1484; 0.9840) |
| R squared | 0.9856 | 0.7341 |
| Slope (*p*-value) | 0.70 (0.00) | 1.12 (0.029) |
| (Lower 95%; Upper 95%) | (0.5876; 0.8251) | (0.5323; 2.2997) |
| Cohen's d | 2.8626 | 5.3896 |
| (Lower 95%; Upper 95%) | (1.0323; 4.6929) | (2.6213; 8.1580) |
| Hedges's g | 2.6424 | 4.9750 |
| (Lower 95%; Upper 95%) | (1.0173; 4.2674) | (2.5724; 7.3777) |

Note:
Elements of the linear models: Dependent variable: CRC prevalence rate; Independent variable_corresponding RSI index; Cohen's d is estimated by calling the cohen.d() function within the Efficient Effect Size Computation or "effsize" package in R software; Hedges' *g* statistics is computed by feeding the instruction (hedges.correction==TRUE) when calling cohen.d().

**Table 5 Results of the nonlinearity tests.**

| Test | Test statistic | *p*-value |
|---|---|---|
| Tsay | 15.32 | 0.00 |
| LR | 110.11 | 0.00 |

CRC prevalence. Moreover, although the effect size magnitude is large for both relationships according to the thresholds provided in *Cohen (1992)*, it is nonetheless significantly smaller for the RSI corresponding to the keyword "coloscopy" than for the RSI sourced for the keyword "colorectal cancer". Consequently, estimations, albeit based on a small sample, agree with previous findings, indicating that online information-seeking for the keyword "colonoscopy" is the best proxy for the prevalence of CRC at the world level.

## Forecasting CRC prevalence from Google search interest

The results of the nonlinearity tests performed on the RSI time series are reported in Table 5. Results agree that the web-query index is nonlinear, as both tests strongly reject the null hypothesis.

Consequently, accounting for the proven existence of data nonlinearity, this study proceeds to estimate a univariate neural network forecasting models that can properly handle this characteristic, *i.e.*, FNNAR. For comparative purposes and to further increase the reliability of results, the widely used ETS and ARIMA models are also estimated, and their respective forecasting capability assessed. Results enclosed in Table 6 reveal that the ANN model, NNAR, is acknowledged as best performing in terms of out-of-sample forecasting accuracy by all scale- and scale-free metrics. It should also be mentioned that the estimated DM test further confirmed the superiority of the neural network model when compared to the second ranked forecasting model, ARIMA.

**Table 6 Accuracy measures over the testing set (out-of-sample) for competing predictive models.**

|       | RMSE  | MAE   | MAPE  | MASE |
|-------|-------|-------|-------|------|
| FNNAR | 14.42 | 9.39  | 17.12 | 2.55 |
| ETS   | 15.38 | 10.77 | 19.65 | 2.92 |
| ARIMA | 14.69 | 10.86 | 17.77 | 2.94 |

**Table 7 Robustness check: accuracy measures over the testing set (out-of-sample) for competing predictive models for the consolidated sample.**

|       | RMSE  | MAE   | MAPE  | MASE |
|-------|-------|-------|-------|------|
| FNNAR | 12.56 | 8.32  | 15.20 | 1.93 |
| ETS   | 20.27 | 17.27 | 28.55 | 4.01 |
| ARIMA | 12.89 | 8.87  | 15.79 | 2.06 |

To assure the robustness of current findings, we first mitigate the sample instability that characterizes GT-sourced RSI series by following the technique proposed by *Medeiros & Pires (2021)*, which solves the sampling instability of RSIs by sourcing and averaging across multiple samples with the same specifications. The aforementioned study shows that this approach improves both model selection and forecast accuracy. Here, we source and average across five different RSI samples constructed by indicating the same keyword, category, and geography. Results, reported in Table 7, again indicate the autoregressive neural network model as best performing in terms of out-of-sample forecasting accuracy and also reveal that employing the consolidated sample does mitigate forecast errors over the testing set for both FNNAR and ETS, offering some support for the findings of *Medeiros & Pires (2021)*.

Moreover, we also perform a secondary robustness check by repartitioning the original RSI sample through the implementation of a 174-55 data splitting rule, as per Box 2.

Box 2. Script for implementing the alternative data partitioning rule

```
RSI <- trends$interest_over_time
x<- ts(RSI$hits)
test_x <- window(x, start=c(175,1),end=c(229,1))#testing window
x <- window(x, end=c(174,1)) #training window
```

Table 8 contains the accuracy measures estimated over the newly defined test-set that were issued by the three competing models, now trained over the first 174 observations in the sample. The nonlinear FNNAR has again been most able to capture variations in data and thus to issue superior out-of-sample forecasts for the RSI.

In light of the consistent superior performance over alternative test sets, the autoregressive NN model is subsequently reestimated over the entire series spanning January 2004 to December 2022 to further issue the expected RSI for CRC over the

**Table 8 Robustness check: accuracy measures over the alternative testing set (*i.e.*, for alternative data partitioning rule) for competing predictive models.**

|        | RMSE  | MAE   | MAPE  | MASE |
|--------|-------|-------|-------|------|
| FNNAR  | 13.14 | 10.31 | 16.62 | 2.45 |
| ETS    | 13.77 | 11.01 | 17.06 | 2.62 |
| ARIMA  | 13.99 | 11.27 | 17.35 | 2.68 |

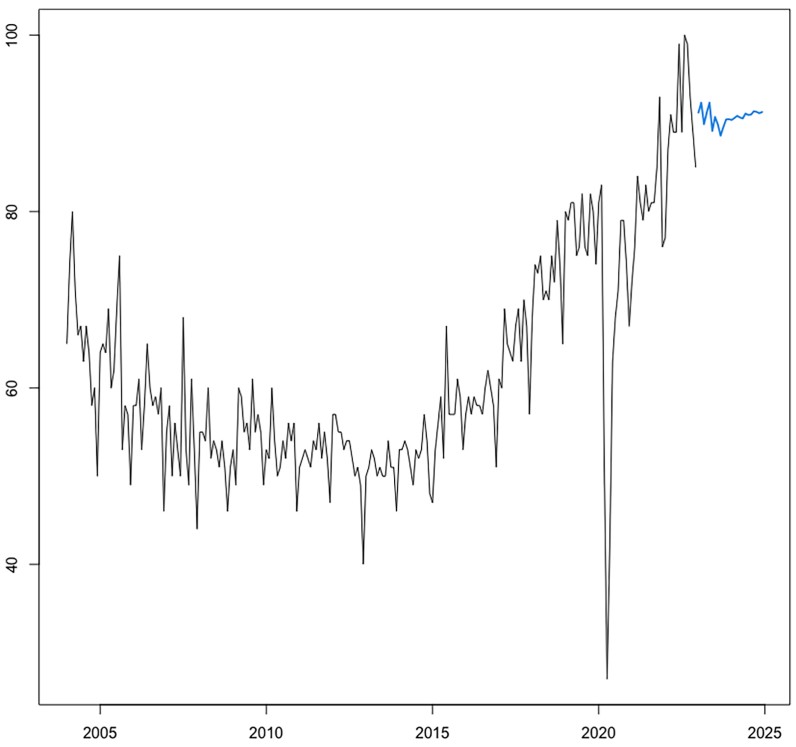

**Figure 10 Forecasting of CRC search interest (RSI) at h = 24.** Source: estimation results. The optimal neural network forecasting models is of the form NNAR (7,4,1), including seven autoregressive lags and four nodes in the hidden layer.

forecasting horizon of 24 months corresponding to the January 2023-December 2024 interval. Projections (visually represented in Fig. 10) show a continuation of population web-search interest in CRC, with an overall growth rate of 16.42 percent relative to the year 2019 (*i.e.*, the point corresponding to the most recent observation for registered CRC prevalence rates at the world level), which in absolute terms would indicate a total burden of disease of 12,361,225 cases, or an additional number of 903,598 CRC "survivors" by the end of 2024. Of note, current projections reinforce the estimations of the Global Cancer Observatory (https://gco.iarc.fr/), reported by *Xi & Xu (2021)* and *Morgan et al. (2023)*, among others, which expect a surge in colorectal cancer incidence and consequently report an expected count of 3.2 million new CRC cases by 2040, or an increase of 63% relative to 2020 levels. Furthermore, current projections also agree with the estimations of the *American Cancer Society (2022a)* cited by the *National Cancer Institute (2022)*, which

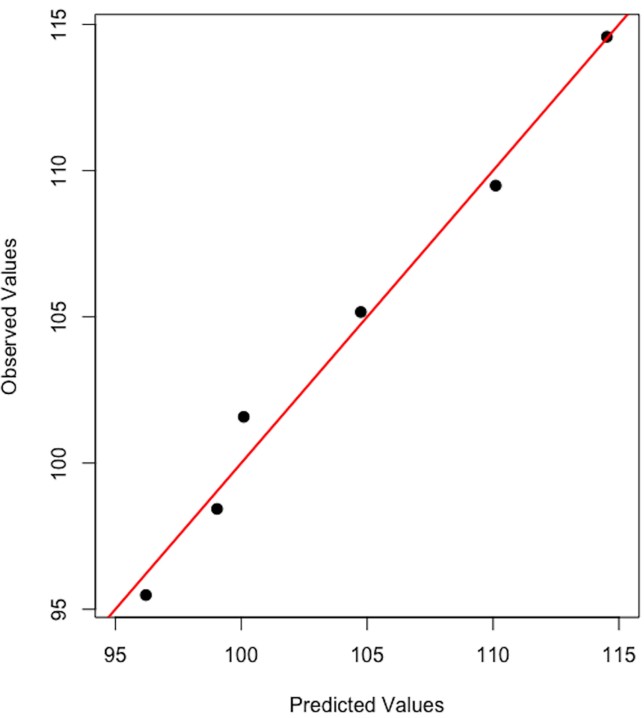

**Figure 11 Observed *vs* predicted global CRC prevalence.**

expect the number of US all-cancers survivors to increase by 24.4%, thus reaching 22.5 million, by 2032.

As a final step, we attempt to validate the capability of the proposed framework by employing it for the task of nowcasting and forecasting the level of worldwide CRC prevalence and compare its projections with the last year of available statistics at the world level (*i.e.*, 2019) provided by the Global Burden of Disease (GBD) database of the Institute for Health Metrics and Evaluation (IHME) (https://www.healthdata.org/gbd).

Figure 11 attests that the linear model constructed within the step 1 of the framework manages to accurately nowcast the level of global CRC prevalence for 2019, indicating a global burden of 11,452,157 compared to 11,457,627 reported by the GBD database of IHME (*i.e.*, 5,470 residual cases at the world level for year 2019).

For forecasting purposes, we subset the first 192 observations in the sample (spanning January 2004–December 2019) and employ an alternative data partitioning strategy (*i.e.*, 180–12). Thus, the FNNAR is trained over the first 180 observations (training window spanning from December 2018) and used to make predictions for the relative search index for "coloscopy" through December 2019. Figure 12 shows the graphical representation of the FNNAR predictions over January to December of 2019, indicating that the model has been able to detect trends fairly well, although it has slightly underestimated the relative search interest over the testing set, issuing an average forecast for RSI of 71.5 over 2019 relative to the actual average RSI level of 78.33. Then, projections issued by FNNAR for the end of 2019 are fed into the linear model constructed during the first step of the framework to finally estimate the global CRC prevalence level corresponding to the year 2019, which is

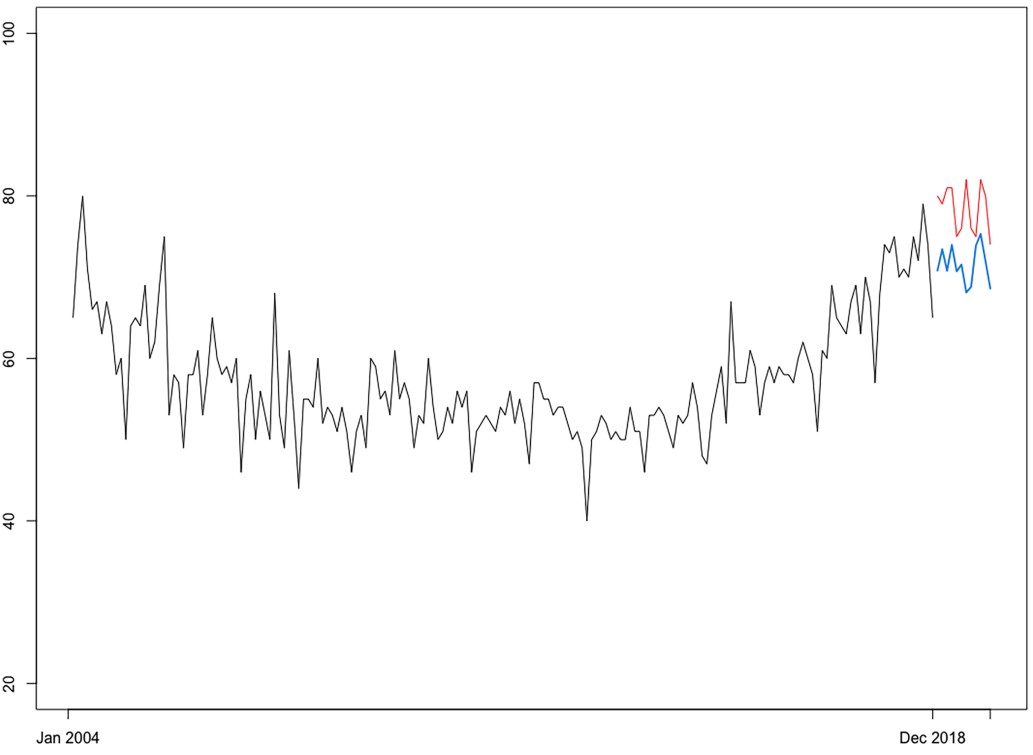

**Figure 12 Projections of CRC search interest (RSI) over January 2019–December 2019 (h = 12) issued by FNNAR (Average of 20 networks, each of which is a 13-7-1 network with 106 weights).** Blue line, NNAR projections; Red line, real RSI index.

then compared to statistics published by IHME. Results show that the proposed framework has been able to capture the global CRC prevalence level for 2019, indicating an interval of (10, 114, 105–11, 812, 163) for the total number of cases globally (*i.e.*, 10,963,134-point forecast), whereas the GBD database of IHME reports a global burden for colorectal cancer of 11,457,627 for 2019. However, to obtain a clearer picture of the framework's ability to project the global burden of the disease, additional validation is required; this will be accomplished with the release of new official statistics corresponding to more recent years.

## DISCUSSION

With the help of big data provided by the Google Trends platform and publicly available data on CRC prevalence rates for 202 countries and several distinct income and geographic panels according to the World Bank classification, this study performs the following tasks: (i) it explores geographical and income-based disparities in CRC prevalence at the world level; (ii) it assesses associations between cancer prevalence and population web-search behavior, thus documenting the relevancy of GT data to proxy for health problem occurrences, and (iii) it comparatively investigates the modeling and forecasting abilities of well-established univariate techniques (including FNNAR, ETS and ARIMA) for RSI time series to finally project CRC prevalence rates for a 24-month forecasting horizon (*i.e.*, up to 2025).

Particularly, based on Google Trends data spanning 2004–2022 and the estimated RSI through the end of 2024, as well as CRC prevalence data for the period 2004–2020, future prevalence rates were projected through the feedforward autoregressive neural network forecasting model (FNNAR), which, due to its ability to learn from the data and deal with non-linearity, emerged as superior in terms of out-of-sample forecasting performance within a pool of three traditional competing univariate predictive models.

CRC prevalence projections point to an increase of 19% relative to the 2019 level, which translates into a total of over 13.64 million people living with colorectal cancer by 2025, or close to 2.2 million over the level registered at the end of 2019, resonating with the findings of previous studies (among others, *Smittenaar et al., 2016*; *Xi & Xu, 2021*) and indicating that increased demands upon the health and insurance services should be expected and carefully planned for. However, similar to *Jakobsen et al. (2021)*, it should be acknowledged that cancer-related indicators should be regularly revised and updated for efficient resource allocation.

Other important results indicate high income and geographical heterogeneity in CRC prevalence at the world level, with high-income countries registering significantly higher prevalence rates than the countries in other income categories, and upper-middle-income countries showing the highest growth rates in CRC prevalence. These findings are in line with *Ades et al. (2013)*, which confirmed that higher economic development and increased expenditures on health at a national level were linked with increased cancer incidence and decreased cancer mortality, which in turn explains the increased prevalence rates encountered in high- and middle-upper-income economies. Current results also agree with previous studies that indicate a rapidly increasing CRC prevalence in less developed countries due to increased exposure to CRC risk factors. Consequently, the study fully supports *Arnold et al. (2017)* in that, to lower the number of CRC patients in the next decades, targeted resource-dependent interventions are required, such as primary prevention in low-income areas and early identification in high-income settings.

The increasing CRC prevalence rates detected, especially in industrialized countries, can be explained by the progress and consequent increase in screening rates, which are in turn reflected in higher cancer incidences. Further, the increase in screening rates has a direct impact on prevalence and contributes, along with lower mortality rates, to the steep increase in prevalence. Indeed, screening seeks to improve patient prognosis by improving early identification and treatment and lowering colorectal cancer (CRC) incidence and death (*Siegel, Miller & Jemal, 2019*; *Gaur & Jagtap, 2022*). Over recent decades, the importance of CRC cancer screening as a main preventive measure has grown (*Colditz & Dart, 2013*), whereas colonoscopy has been recognized as the gold standard for CRC screening (*Food and Drug Administration (FDA), 2021*). For example, at the United States (US) level, the state with the highest rate of screening has been identified as Massachusetts (*Colditz & Dart, 2013*).

Additionally, the relevancy of web-query data as a proxy for awareness and interest in cancer screening has been previously documented (*Schootman et al., 2015*). As such, the strong connection between RSI for "colonoscopy" and the CRC prevalence rate detected in the current study can be explained through the screening-incidence-prevalence

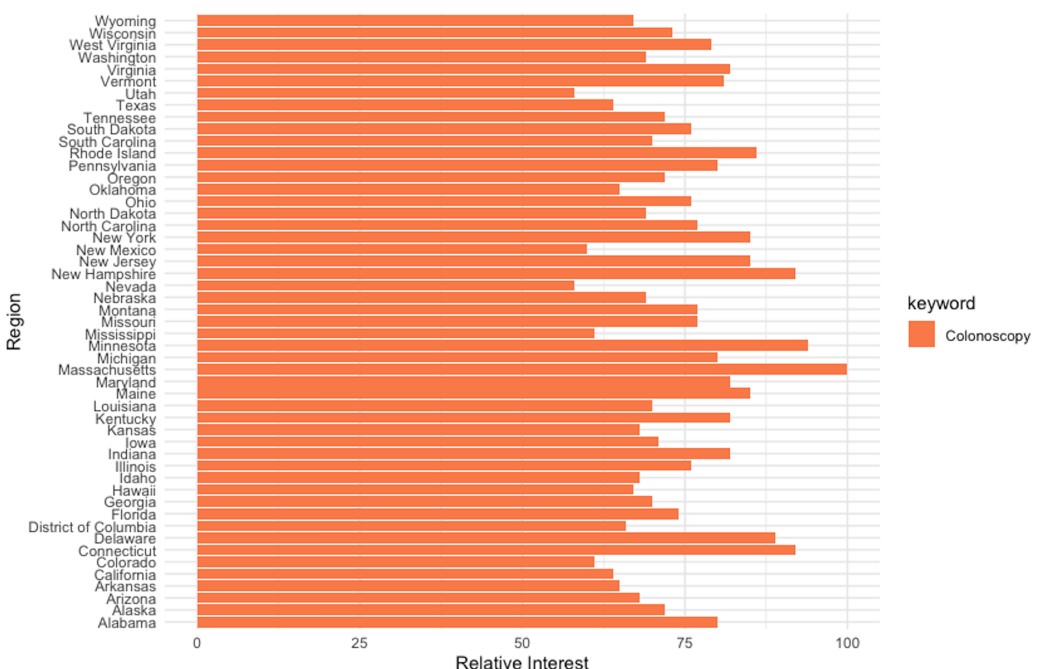

**Figure 13** **Interest-by-region analysis for the keyword "colonoscopy" at the US level.** Source, Authors' representation in R software.

transmission channel. Furthermore, the breakdown by region of the web-search interest for the keyword "colonoscopy" performed at the "US" geography level (enclosed in Fig. 13) identifies Massachusetts as the US state with the highest relative search interest, which further supports on one hand the aforementioned argumentation and on the other hand the previous literature on the use of web query data in health research.

Overall, this research resonates with *Jun, Yoo & Choi (2018)* that the use of big data, such as the GT relative search volume indexes, has shifted recently from modeling to forecasting. Additionally, it contends that, for cancer research, web-search data might present several advantages over official statistics, including: (i) first, official data are reported with a lag, and thus timely accurate research is hampered by this unavoidable delay; (ii) second, outlier events that disrupt health care systems, such as the COVID-19 pandemic, cause significant underdiagnosis (*Jacob et al., 2021*; *Marques et al., 2021*), which invalidates official data that no longer accurately capture the variation in incidence, hence also leading to faulty prevalence indicators. In line with *Salathé et al. (2012)* and *Sulyok, Ferenci & Walker (2021)*, local and timely information on disease and health patterns is extracted from population web-search interests. Moreover, future research focused on the most vulnerable countries, with undeveloped healthcare systems and often unreliable or nonexistent statistics, can draw significant benefits from extracting reliable health information from web-queries.

## Limitations

Research that employs Google Trends data, as is the case with the current study, is based on the central premise that people turn to Google to find subject-related information when

they need it, and, due to its visibility, this information demand can subsequently be used as a reliable predictor in a variety of settings (*Bleher & Dimpfl, 2022*). Moreover, compared to official statistics, GT data are available in real-time and cover a wide sample of countries, regions, and even some large cities, offering valuable insights especially when other data are lacking or are only available with important time lags (*Eichenauer et al., 2022*), as is the case with cancer prevalence data that makes the subject of the current research. Consequently, Google Trends data have proven particularly helpful in shedding light on population health behaviors (*Arora, McKee & Stuckler, 2019*) and hold the potential to improve the capabilities of public health surveillance systems (*Althouse et al., 2015*; *Tkachenko et al., 2017*). However, there are also some non-trivial limitations to the GT data that should be properly acknowledged (*Tudor, 2022a*) and, where possible, overcome.

First, Google Trends RSI is aggregated from a sample of search queries (*Narita & Yin, 2018*), thus suffering from sample bias (*Medeiros & Pires, 2021*) which, together with sample instability that arises from its construction (*Eichenauer et al., 2022*), can have serious consequences for the reliability of research findings that use GT data. In fact, this makes any study employing GT data inherently irreproducible (*Rovetta, 2021*). To mitigate, albeit not overcome, these issues, we construct a consistent sample by additionally employing a robust sampling procedure that involves sourcing multiple samples at various intervals and averaging across to construct a "consolidated" RSI. Moreover, it should also be mentioned that the aforementioned weaknesses related to GT data are prominent for less searched topics or keywords (*Medeiros & Pires, 2021*), which was not the case for the main keyword of interest in this study (*i.e.*, "colonoscopy") over the analyzed period. Moreover, previous research has shown that "worldwide" RSIs, as is the case in the current study, are the most reliable GT samples (*Rovetta, 2021*).

Second, Google Trends data is susceptible to being influenced by external events, such as news events or media coverage (*Sato et al., 2021*; *Satpathy, Kumar & Prasad, 2023*). Thus, given that the period under consideration in the current study contains the COVID-19 pandemic, it should be noted that the media coverage may have influenced the search volume in pandemic-related terms during the analyzed period (*Rovetta & Castaldo, 2021*), and, given the construction of the RSI as explained in the method section, this in turn may have direct consequences for the relative interest in unrelated terms, such as the one used in this research (*i.e.*, "colonoscopy"). Nonetheless, it should also be mentioned that even allowing for the possibility that media coverage may affect population web searches, GT data still offers a way to measure web interest in a particular subject more effectively than any other approaches previously employed (*Rovetta, 2021*). Furthermore, as the COVID-19 pandemic significantly affected CRC screening rates, with direct consequences for the accuracy of CRC prevalence data and their usefulness for cancer research, we align with previous studies and argue that digital traces are the best available tools for assessing disease prevalence.

## CONCLUSIONS

Significant advancements in cancer screening and treatment have increased both cancer incidence statistics and patients' life expectancy. As a result, a larger portion of the

population is becoming increasingly reliant on social services, particularly health care, and this trend is especially visible in high- and upper-middle-income countries. In this context, the prevalence of cancer has emerged as a vital indicator for effective policy and proper resource allocation, both of which require reliable estimates. Consequently, accurate CRC prevalence projections carry important implications, informing the policymakers, the insurance companies, and the public authorities on the estimated future burden, the size of programs aimed at releasing this burden, and the related research financing needs.

However, unlike cancer incidence and mortality rates, cancer prevalence projections are not routinely issued by national and international agencies, and the limited or nonexistent cancer statistics for large portions of the world, along with the high heterogeneity among world nations, further complicate the task of producing timely and accurate CRC projections. In this context, population interest in the form of internet-submitted queries are crucial for enhancing cancer statistics and ultimately assisting cancer research.

This article proposes a three-step framework that constructs a CRC prevalence model from Google Trends search volume series to forecast CRC prevalence at a monthly frequency with a 24-month forecasting horizon. Multiple ETS, ARIMA, and FNN models are fitted to the monthly Google Trends series, and all best-fit models in each category are employed for the task of issuing 45-point forecasts on the test data window and subsequently ranked by multiple accuracy metrics. Results consistently indicate that the machine-learning neural network outperforms in the out-of-sample setting, due to its capability to capture non-linearity, which has been identified in the data by two alternative preliminary tests.

Other important findings document that population web-search interest is a reliable indicator of disease prevalence and a linear model that links RSI and CRC prevalence at the world level can accurately nowcast the global burden of disease.

Estimates issued through the proposed framework for a 24-month forecasting horizon indicate that the global burden of colorectal cancer (CRC) can increase by over 10.6%, from about 11.46 million people living with CRC at the end of 2019 to more than 12.67 million cases by 2025.

Of note, the significant geographical and income-based disparity in CRC prevalence is documented on the global scene, with countries that belong to the East Asia and Pacific region and the upper-middle-income panel being most vulnerable to this burden. This finding in turn carries important implications and should be acknowledged in the process of issuing equitable global health policies.

Moreover, as the costs connected with CRC are extremely high for both individuals and society, current findings further emphasize the stringent need to mitigate this global burden through a focus on prevention and early detection. In this context, public policies aimed at increasing health expenditures and subsequent CRC screening, corroborated with public campaigns directed at educating the population to decrease exposure to the main modifiable (*i.e.*, non-genetic) risk factors such as alcohol consumption, tobacco use, overweight and obesity, poor dietary habits, a lack of regular physical activity, *etc.*, are paramount. In addition, the findings suggest that public health authorities should take

measures to increase cancer screening rates during pandemics, which would have positive externalities such as reducing the global burden and enhancing official statistics.

However, it is important to underline that projections should be revised regularly for healthcare planning and resource allocation purposes. Additionally, another limitation of the proposed framework should be acknowledged, as it does not include any further progress in treatment and/or improvement in screening rates over the forecasting horizon (*i.e.*, the framework only operates under the "*status quo*" hypothesis).

### Funding
The authors received no funding for this work.

### Competing Interests
The authors declare that they have no competing interests.

### Author Contributions
- Cristiana Tudor conceived and designed the experiments, performed the experiments, analyzed the data, performed the computation work, prepared figures and/or tables, authored or reviewed drafts of the article, and approved the final draft.
- Robert Aurelian Sova conceived and designed the experiments, performed the experiments, authored or reviewed drafts of the article, and approved the final draft.

### Data Availability
Code and raw data are available in the Supplemental Files.

### Supplemental Information
Supplemental information for this article can be found online at http://dx.doi.org/10.7717/peerj-cs.1518#supplemental-information.

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
