# Peer review of "Mining Google Trends data for nowcasting and forecasting colorectal cancer (CRC) prevalence"

_PeerJ Computer Science, doi:10.7717/peerj-cs.1518_

## Round 0.1 · original submission · Major Revisions

Please address the reviewer comments in a point-to-point manner.

Reviewer 1 ·

Basic reporting

The article discusses the lack of cancer prevalence projections for colorectal cancer (CRC), despite it being the third most prevalent and second most lethal cancer in the world. The article proposes using population web-search interest, as shown by Google Trends (GT) data, to forecast CRC prevalence. The article employs a three-step framework using statistical and machine-learning models to forecast CRC prevalence up to 2025. Results show that the number of people with colorectal cancer will continue to rise over the next 24 months, emphasizing the urgency for public policies aimed at reducing modifiable risk factors. Additionally, the article suggests that public health authorities should implement measures to increase cancer screening rates during pandemics to improve cancer statistics and mitigate the global burden.

Experimental design

Here are some suggestions to improve the experimental design:

Include a validation step: To ensure that the Google Trends data accurately reflects the true prevalence of cancer, the study could validate the data against existing cancer prevalence data. This would help establish the reliability and validity of the Google Trends data as a proxy for cancer prevalence.

Increase the multiple search terms: Instead of relying on a single search term, the study could use multiple search terms to capture a broader range of web-search interests related to colorectal cancer.

Use a representative sample: To improve the generalizability of the findings, the study could use a representative sample of populations from different regions, ages, genders, and socioeconomic backgrounds.

Consider different modeling techniques: The study could use a variety of statistical and machine learning models to assess the robustness of the findings. This could include models that take into account different variables, such as demographic data, lifestyle factors, and environmental risk factors.

Validity of the findings

Here are some suggestions to improve the validity of the findings:

Use multiple data sources: To improve the validity of the findings, the study could use multiple data sources to validate the results obtained from Google Trends data. This could include existing cancer registry data or data from other health information systems.

Consider using a longitudinal design: A longitudinal design would allow the study to track changes in web-search interest over time and to establish a temporal relationship between web-search interest and cancer prevalence.

Control for confounding variables: To isolate the effect of web-search interest on cancer prevalence, the study should control for potential confounding variables, such as demographic data, lifestyle factors, and environmental risk factors.

Validate the findings with external data: To improve the external validity of the findings, the study could validate the results with external data from other studies or sources. This would help to establish the generalizability of the findings and provide additional support for the use of Google Trends data in cancer prevalence projections.

Additional comments

I would like to highlight the pros of the paper:

Novel approach: The paper proposes a novel approach to forecasting colorectal cancer prevalence at the global level using Google Trends data and various statistical and machine-learning models.

Practical implications: The paper's findings have practical implications for public policy and health authorities, emphasizing the urgency for reducing population exposure to modifiable risk factors and implementing measures to increase cancer screening rates during pandemics.

Data analysis: The paper performs a thorough data analysis, including data mining to evaluate the relevance of Google Trends data as a surrogate for the number of CRC survivors and training and testing various statistical and machine-learning models.

Forecasts: The paper generates point forecasts up to 2025 using the best forecasting model in terms of out-of-sample forecasting ability.

Contribution to cancer research: The paper's approach and findings may contribute to improving cancer statistics and, in the long run, cancer research.

Transparency: The paper provides a transparent description of its methodology, including data sources and statistical models used.

·

Basic reporting

I point out that all the references reported in my review have the sole scope of justifying my statements and providing data and help to the authors. My final assessment is in no way
dependent on their inclusion in the manuscript.

Major comments

1) The many limitations and issues of Google Trends should be properly highlighted to provide a neutral and unbiased background. Indeed, many publications show that web searches are strongly influenced by mass media coverage (e.g., https://pubmed.ncbi.nlm.nih.gov/34842858/, https://pubmed.ncbi.nlm.nih.gov/28756828/, https://bmcmedresmethodol.biomedcentral.com/articles/10.1186/s12874-021-01338-2, https://www.ncbi.nlm.nih.gov/labs/pmc/articles/PMC8460424/). Moreover, Google Trends data may be subject to marked and unpredictable fluctuations and anomalies (e.g., https://pubmed.ncbi.nlm.nih.gov/34113751/).

=============

Minor comments

m1) I suggest avoiding anticipating the paper's findings in the introduction since this section should only provide background on the topic.

Experimental design

Major comments

2) Lines 187-188. How were the confidence intervals calculated? Have the assumptions been tested for their representative validity? If yes, how?

3) Lines 187-188. Generally, confidence intervals are not the best indicators for data heterogeneity (e.g., variance is more appropriate for this scope https://pubmed.ncbi.nlm.nih.gov/36751163/).

4) The limitations discussed in the suggested references in my comment "1" should be examined properly, as they can impair the manuscript validity.

5) How was the reliability and appropriateness of all the models, tests, and methods verified? This aspect could be explained in detail in a supplementary file.

Validity of the findings

Major comments

6) I suggest providing confidence intervals for all the results (e.g., correlation). This is essential to understand the effect size. Measures like Cohen's D could also be considered in this scope.

7) P-values should be used, at best, as graded measures of data compatibility with the test hypothesis (https://pubmed.ncbi.nlm.nih.gov/28698825/, https://www.ncbi.nlm.nih.gov/pmc/articles/PMC4877414/). Therefore, where required, I suggest showing the exact P-values (unless P<.001 or similar). Moreover, I suggest talking of degrees of evidence instead of using thresholds (e.g., “low, medium, high significance” instead of “significant” and “non-significant”). For example, P=.049 and P=.051 are always similar results. In this regard, S-values could be more understandable for the reader.

8) P-values are reliable only and only if all the tests' assumptions have been properly verified (https://pubmed.ncbi.nlm.nih.gov/36751163/). Therefore, it is fundamental to address the points raised in my comments 2 and 5.

Additional comments

Major comments

9) The concepts of effect size and statistical significance should be rethought and discussed through the manuscript based on the above comments, since they represent two different aspects.

10) As verifiable in the "Interest Over Time" section of Google Trends, the data collection systems were subject to various changes during the period 2004-2022. Were the impact of these changes considered in this paper?

11) The paper needs a limitation section.

Reviewer 3 ·

Basic reporting

Figures and tables are correctly displayed.
The dataset and code were opened and worked correctly, as well as being described in English.

Experimental design

Of course, the research paper falls within the aims and scope of the journal, and the methods are described with sufficient detail and information.

Validity of the findings

All underlying data have been provided; they are robust, statistically sound, and controlled, and the conclusions are well stated, linked to the original research question, and limited to supporting results.

Additional comments

I have no comments because the title and abstract are very clear.

---

## Round 0.2 · accepted · Accept

The authors have addressed the 3 reviewers' comments.

Reviewer 1 ·

Basic reporting

The authors have responded adequately and substantially improved the quality of the paper. I do not have much more to contribute.

Experimental design

The authors have responded adequately and substantially improved the quality of the paper. I do not have much more to contribute.

Validity of the findings

The authors have responded adequately and substantially improved the quality of the paper. I do not have much more to contribute.

Additional comments

The authors have responded adequately and substantially improved the quality of the paper. I do not have much more to contribute.

·

Basic reporting

The authors addressed my concerns properly.

Experimental design

Everything is now detailed and reproducible (as much as possible).

Validity of the findings

Although the paper has limitations, these are now well explained and the work is scientifically sounding. Therefore, if contextualized, this paper can contribute to the scientific literature about infodemiology.

Additional comments

I wish the authors the best for this and their future projects.

Reviewer 3 ·

Basic reporting

Figures and tables are correctly displayed.
The dataset and code were opened and worked correctly, as well as being described in English.

Experimental design

The research paper falls within the aims and scope of the journal, and the methods are described with sufficient details and information.

Validity of the findings

All underlying data have been provided; they are robust, statistically sound, and controlled, and the conclusions are well stated, linked to the original research question, and limited to supporting results.

Additional comments

No comment due to the title and abstract being clear.